# More than just availability: Who has access and who administers take-home naloxone in Baltimore, MD

**Lauren Dayton**[1]*, **Rachel E. Gicquelais**[2], **Karin Tobin**[1], **Melissa Davey-Rothwell**[1], **Oluwaseun Falade-Nwulia**[3], **Xiangrong Kong**[4], **Michael Fingerhood**[5,6], **Abenaa A. Jones**[1], **Carl Latkin**[1]

1 Department of Health, Behavior and Society, Johns Hopkins Bloomberg School of Public Health, Baltimore, Maryland, United States of America, 2 Department of Epidemiology, Johns Hopkins Bloomberg School of Public Health, Baltimore, Maryland, United States of America, 3 Division of Infectious Diseases, Department of Medicine, Johns Hopkins University School of Medicine, Baltimore, Maryland, United States of America, 4 Department of Ophthalmology, Johns Hopkins University School of Medicine, Baltimore, Maryland, United States of America, 5 Department of Medicine, Johns Hopkins University School of Medicine, Baltimore, Maryland, United States of America, 6 Department of Mental Health, Johns Hopkins Bloomberg School of Public Health, Baltimore, Maryland, United States of America

* ldayton2@jhu.edu

**Data Availability Statement:** All relevant data are within the manuscript and its Supporting Information files.

## Abstract

### Background

Fatal opioid overdose is a pressing public health concern in the United States. Addressing barriers and augmenting facilitators to take-home naloxone (THN) access and administration could expand program reach in preventing fatal overdoses.

### Methods

THN access (i.e., being prescribed or receiving THN) was assessed in a Baltimore, Maryland-based sample of 577 people who use opioids (PWUO) and had a history of injecting drugs. A sub-analysis examined correlates of THN administration among those with THN access and who witnessed an overdose (N = 345). Logistic generalized estimating equations with robust standard errors were used to identify facilitators and barriers to accessing and using THN.

### Results

The majority of PWUO (66%) reported THN access. In the multivariable model, decreased THN access was associated with the fear that a person may become aggressive after being revived with THN (aOR: 0.55, 95% CI: 0.35–0.85), police threaten people at an overdose event (aOR: 0.68, 95% CI: 0.36–1.00), and insufficient overdose training (aOR: 0.43, 95% CI: 0.28–0.68). Enrollment in medication-assisted treatment, personally experiencing an overdose, and graduating from high school were associated with higher access. About half (49%) of PWUO with THN access and who had witnessed an overdose reported having administered THN. THN use was positively associated with "often" or "always" carrying THN

**Funding:** National Institutes of Health grants DA022961(CL), DA040488(CL), AI102623(REG), and K23DA041294(OFN) supported this research. The funders had no role in study design, data collection and analysis, decision to publish, or preparation of the manuscript.

**Competing interests:** The authors have declared that no competing interests exist.

(aOR: 3.47, 95% CI: 1.99–6.06), witnessing more overdoses (aOR:5.18, 95% CI: 2.22–12.07), experiencing recent homelessness, and injecting in the past year. THN use was reduced among participants who did not feel that they had sufficient overdose training (aOR: 0.56, 95% CI: 0.32–0.96).

## Conclusion

THN programs must bolster confidence in administering THN and address barriers to use, such as fear of a THN recipient becoming aggressive. Normative change around carrying THN is an important component in an overdose prevention strategy.

## Introduction

Opioids were involved in more than 47,000 deaths in the US in 2017, killing more people than motor vehicle accidents and firearms [1–3]. Recently, the majority of fatal opioid overdose deaths have been attributed to fentanyl, a synthetic opioid [1,4]. The proliferation of fentanyl and its analogs in recent years has increased the risk of overdose fatality due to their heightened potency [5]. Fatal opioid overdose is preventable with the administration of opioid overdose reversal drugs, such as naloxone. Naloxone is a fast-acting μ-opioid receptor antagonist that competitively displaces opioids, reversing the central nervous system depression that occurs during an opioid overdose [6]. The World Health Organization guidelines strongly recommend equipping people who are likely to witness an overdose with naloxone rescue kits ("take-home naloxone", THN) and providing training in the management of opioid overdose [7]. The efficacy of naloxone in reversing an opioid overdose is largely independent of route of administration, and THN formulations include intramuscular, subcutaneous, and intranasal [7]. There are many THN training programs available, however the core component of all programs is to enable the management of an opioid overdose through effective administration of THN [7]. People who use opioids (PWUO) are especially in need of THN as they are highly likely to witness overdose events [8,9]. Strang and colleagues found that 97% of PWUO report having witnessed an overdose and Ogeil and colleagues found that 21% of prescription overdose deaths were witnessed by another person [10,11].

In efforts to increase access to THN, many municipalities have implemented interventions such as provision of THN at pharmacies and other community-based sites providing services for individuals with substance use disorders [12–14]. For example, in Baltimore City, there has been a standing order policy in place since 2015 which eliminates the need for individual prescriptions for THN [15]. Despite concentrated efforts to increase THN distribution and training, many PWUO have never received THN and overdose response strategies by bystanders are often inconsistent and ineffective [10,16].

Few studies have examined barriers and facilitators of THN access and administration among PWUO [17–19]. Kenney et al. examined correlates of THN use among PWUO using a sample of participants who were enrolled in an inpatient opioid detoxification program [18]. The study found that recent injection drug use, history of overdose, witnessing an overdose in the past year, non-Black race, and detoxification from heroin were associated with THN administration [18]. Perceived negative consequences of administering THN may also reduce access to and use of THN. Among people enrolled in medication-assisted treatment (MAT), Khatiwoda and colleagues found that fear of prosecution by police was cited as a reason for not having access to THN [19]. Further, a recent qualitative study identified that some PWUO

report not wanting to use THN for fear of an aggressive response from the overdose victim after administering THN [17]. While THN has minimal side effects, it does induce acute withdrawal symptoms in opioid-dependent individuals [6,20], and precipitated withdrawal can provoke an aggressive reaction in some individuals [20–22].

This fear of police and an aggressive response by the overdose victim after administering THN may prevent accessing and using THN during a witnessed overdose. Additional, yet unexplored barriers to access and use of THN may include insufficient training to effectively administer THN and inconsistencies in carrying THN. Further, access to health services may increase training on THN and facilitate access and use of THN [13]. The primary aim of the current study was to examine facilitators and barriers to THN access among PWUO. The secondary aim was to identify predictors of administering THN among PWUO who had witnessed an overdose and had access to THN.

## Methods

### Study participants

Study participants were recruited in Baltimore, Maryland, for a randomized clinical trial of an intervention to enhance Hepatitis C and HIV prevention and care among people using substances and residing in impoverished neighborhoods. A total of 518 index participants were recruited using street-based outreach, advertisement, and word-of-mouth. Inclusion criteria for the baseline screening visit included being 18 years of age or older and a history of lifetime injection drug use. Index participants were encouraged to recruit network members who were drug using and/or sexual partners. The study sample includes an additional 71 network members recruited by index participants who completed the baseline screening visit. The current analysis was restricted to 577 of 589 participants (98%), 511 index and 66 network members, who completed a baseline screening visit between December 2016 and September 2018 and reported ever using illicit opioids or non-medical prescription opioids. All participants provided written informed consent. Trained study staff administered a face-to-face survey to collect participant demographic and risk factor data. Sensitive risk behavior questions were assessed via audio computer assisted self-interviewing (ACASI). Participants were paid 20 US dollars for completing the survey. Study protocols were approved by Johns Hopkins Bloomberg School of Public Health IRB.

### Measures

**Outcome: Access to and administration of THN.**   Our primary outcomes were access to and administration of THN. THN access was assessed by asking participants, "Have you ever been prescribed or received a kit containing Narcan?" Prior to this question, participants were informed that "Narcan/naloxone is a prescription drug that can be administered to reverse an opiate overdose." THN administration during a witnessed overdose was assessed using the question, "have you ever used Narcan to reverse an opiate overdose?" Use of THN to respond to a witnessed overdose was only examined among the subset of PWUO who reported receiving THN and who witnessed ≥1 overdoses during their lifetime.

**Socio-demographics.**   Socio-demographic variables included measures of age, gender, education, and homelessness. Age was categorized based on quartiles. Gender was analyzed as a dichotomous measure comparing male to female. Education was defined as a binary measure comparing not graduating from high school vs. high school graduation or higher educational attainment. Homelessness was self-reported as experiencing homelessness at any time in the past 6 months.

**THN availability.** The frequency of carrying THN was assessed through the question, "how often do you carry Narcan with you?" For the analysis, the response categories were dichotomized as "never," "rarely," or "sometimes" versus "often" or "always."

**Drug use and overdose characteristics.** Injection drug use in the past year was assessed through the question, "When was the last time you injected drugs to get high?" One person did not answer the question and was coded at the median of having injected in the past year. Personal overdose experience was assessed as responding with one or more to the question, "How many times in your life have you overdosed?" The number of overdoses witnessed was categorized into five categories using natural breaks in the distribution (0, 1–2, 3–4, 5–9, or ≥10 overdoses).

**Fentanyl perceptions.** Perceived prevalence of fentanyl was assessed by the question "What percentage of heroin on the streets of Baltimore do you think contains fentanyl?" Responses were categorized as "more than half" versus "half," "less than half," or unsure/not familiar with fentanyl.

**Perceived barriers of responding.** Three items assessed perceived barriers to assist in the event of an overdose. The first assessed insufficient training to respond to an overdose with the statement, "I am going to need more training before I would feel confident to help someone who has overdosed." For the analysis, the variable was dichotomized as "strongly agree" or "agree" versus "strongly disagree," "disagree," or "neither agree nor disagree." The second assessed fear of an aggressive response after administration of THN using the statement, "I would be afraid of giving Narcan in case the person becomes aggressive afterward." We dichotomized responses as "strongly agree" or "agree" versus "strongly disagree," "disagree," or "neither agree nor disagree" for the analysis. The third assessed the perceived threat from police with the question, "When the police show up at an overdose, how often do they threaten the people present, including the victim, with drug charges or arrest?" Responses were dichotomized as "never" or "rarely" versus "sometimes," "often," or "always." One participant responded, "don't know" and was categorized as "never."

**Health services access and engagement.** Access to health services was assessed with survey items on receiving MAT and having health insurance. MAT was defined as reporting currently taking buprenorphine, methadone, or naltrexone. Health insurance status was dichotomized as having health insurance versus not having health insurance at the time of the survey.

**Location of THN training.** Among participants who reported having been trained in THN, training location was assessed through the open-ended question: "If you wanted to get Narcan or a refill where would you go to obtain it?" Responses were assessed for recurring locations and coded as community outreach programs (e.g. syringe service programs and community events), recovery programs, clinics, detention centers, through friends, or other avenues.

## Statistical analysis

Differences in the distribution of participant characteristics by the primary (access to THN) and secondary (THN administration) outcomes were accessed with Chi-square tests. Two multivariable models were constructed to identify independent predictors of access to THN and administration of THN. Due to anticipated correlations among some of the variables, backwards stepwise selection was used to refine multivariable models with a threshold p-value of .10. The demographic variables of age, gender, and education were also included in the multivariable models. Logistic generalized estimating equations with robust standard errors accounted for the clustered structure of the data (i.e., that index participants recruited network

members). A sensitivity analysis was performed among PWUO who injected drugs in the past year, as they represent a subsample at elevated risk of overdose and in need of THN. Statistical analysis was conducted with STATA version 14 software [23].

## Results

The mean age of study participants (N = 577) was 47 years (SD: 11). The majority were male (66%), had graduated from high school (61%) reported injecting drugs in the past year (63%), were currently enrolled in MAT (63%) and had health insurance coverage (92%). Homelessness in the past 6 months was reported by 47% of study participants. The majority had witnessed (87%) and personally experienced (66%) an overdose in their lifetime. Fentanyl was perceived to be in more than half the heroin supply by 57% of participants, and 57% reported having insufficient training to respond to an overdose. About one-quarter (26%) feared an aggressive response after giving THN, and 51% perceived that police would threaten people at the scene of an overdose with drug charges or arrest.

Additionally, an analysis of study participants who stated that they had been trained to use THN (n = 338) indicated that participants were primarily trained through community outreach programs (36%), recovery programs (34%), and clinics (27%). The minority of participants reported being trained through detention centers (2%), through friends (1%), or other avenues (1%).

### Correlates of access to THN

Two-thirds (66%) of study participants reported access to THN. The bivariate analysis compared PWUO with access to THN to those without access (Table 1; N = 577). In the univariate analysis, access to THN was associated with younger age, completion of high school, recent injection drug use, personal overdose experience, witnessing more overdoses, perception of higher amounts of fentanyl in heroin, and current enrollment in MAT. Perceptions of insufficient training and that a THN recipient would become aggressive was associated with decreased access to THN. In the multivariable model (Table 2), access to THN was positively associated with having completed high school education or above (adjusted odds ratio, aOR: 1.68, 95% CI: 1.13–2.50), personally experiencing an overdose (aOR: 2.68, 95% CI:1.79–4.01), and currently receiving MAT (aOR: 3.87, 95% CI: 2.59–5.78). Reduced THN access was associated with a perception of needing more overdose response training (aOR: 0.43, 95% CI: 0.28–0.68), fearfulness that a person would become aggressive after being revived with THN (aOR: 0.55, 95% CI: 0.35–0.85), and perception that police would threaten people at an overdose with charges or arrest (aOR: 0.68, 95% CI: 0.46–1.00). Age, recent injection drug use, number of witnessed overdoses, and perceptions of fentanyl in the heroin supply did not remain independent predictors of access to THN in the multivariate model.

A sensitivity analysis among the subset of participants who reported injecting drugs in the past year (n = 363) was highly consistent with findings among all participants (results not shown). However, education was no longer a significant predictor of access to THN.

### Correlates of THN administration

Of the 380 participants with access to THN, 345 reported witnessing an overdose and were included in an examination of correlates of using THN. About half of these participants (49%) reported ever administering THN. Bivariate analyses (Table 1) found that THN use was higher among people who had witnessed an overdose, were younger, more frequently carried THN, experienced homelessness, recently injected drugs, witnessed more overdoses, and believed there was more fentanyl in the heroin supply. Perceptions of insufficient overdose training

**Table 1. Bivariate correlates of access and administration of take-home naloxone (THN) among people who use opioids in Baltimore, MD, USA.**

| | THN Access | | | THN Administration | | |
|---|---|---|---|---|---|---|
| | No (n = 197) | Yes (n = 380) | P value | No (n = 177) | Yes (n = 168) | P value |
| **Socio-Demographic Characteristics** | | | | | | |
| Age (years) | | | | | | |
| 40–49 | 47 (23%) | 94 (25%) | 0.003 | 50 (28%) | 35 (21%) | <0.001 |
| 50–55 | 48 (24%) | 97 (25%) | | 49 (28%) | 36 (21%) | |
| 56+ | 61 (31%) | 70 (18%) | | 41 (23%) | 24 (14%) | |
| Male | 136 (69%) | 244 (64%) | 0.246 | 110 (62%) | 114 (68%) | 0.267 |
| High school/GED+ | 106 (54%) | 248 (65%) | 0.007 | 107 (60%) | 115 (68%) | 0.121 |
| Homeless (past 6 mos.) | 83 (42%) | 187 (49%) | 0.106 | 69 (39%) | 105 (63%) | <0.001 |
| **THN Availability** | | | | | | |
| Carry THN Often/Always | --- | --- | | 30 (17%) | 77 (46%) | <0.001 |
| **Drug Use & Overdose History** | | | | | | |
| Injected (past 12 mos.) | 107 (54%) | 256 (67%) | 0.002 | 100 (56%) | 136 (81%) | <0.001 |
| Ever overdosed | 99 (50%) | 281 (74%) | <0.001 | 131 (74%) | 131 (78%) | 0.389 |
| No. of overdoses witnessed | | | | | | |
| 0 | 38 (19%) | 35 (9%) | 0.001 | --- | --- | |
| 1 to 2 | 52 (27%) | 78 (20%) | | 58 (33%) | 20 (12%) | <0.001 |
| 3 to 4 | 41 (21%) | 94 (25%) | | 55 (31%) | 39 (23%) | |
| 5 to 9 | 28 (14%) | 67 (18%) | | 28 (16%) | 39 (23%) | |
| 10 + | 38 (19%) | 106 (28%) | | 36 (20%) | 70 (42%) | |
| **Perceptions of Fentanyl** | | | | | | |
| >50% of heroin supply has fentanyl | 99 (50%) | 228 (60%) | 0.025 | 97 (55%) | 112 (67%) | 0.024 |
| **Perceived Barriers of Responding** | | | | | | |
| Perception of insufficient overdose training | 143 (73%) | 185 (49%) | <0.001 | 105 (59%) | 58 (35%) | <0.001 |
| Perception that Narcan recipient will become aggressive | 74 (38%) | 77 (20%) | <0.001 | 48 (27%) | 22 (13%) | 0.001 |
| Perception that police will threaten people at overdose scene | 109 (55%) | 184 (48%) | 0.115 | 82 (46%) | 85 (51%) | 0.428 |
| **Health Services Access and Engagement** | | | | | | |
| Medication-Assisted Treatment | 82 (42%) | 281 (74%) | <0.001 | 135 (76%) | 120 (71%) | 0.306 |
| Currently have health insurance coverage | 175 (89%) | 354 (93%) | 0.074 | 167 (94%) | 154 (92%) | 0.327 |

---variable not included in the model

and that THN recipient would become aggressive were associated with not administering THN. In the multivariable model (Table 2), THN availability (i.e. often/always carrying THN) (aOR: 3.47, 95% CI: 1.99–6.06) and homelessness (aOR:1.82, 95% CI:1.08–3.07) were associated with higher odds of using THN. Both injecting in the past year (aOR:2.61, 95% CI:1.43–4.76) and witnessing more overdoses were positively associated with administering THN. Witnessing 5–9 overdoses was associated with 5.18-fold higher odds of administering THN compared to witnessing one or two (aOR:5.18, 95% CI:2.22–12.07). Administration of THN was reduced among those who felt neutral or agreed about needing further overdose training (aOR: 0.56, 95% CI:0.32–0.96). Fear that a THN recipient would become aggressive was marginally associated with reduced odds of administering THN (aOR: 0.54, 95% CI:0.28–1.04). Perceived prevalence of fentanyl was not found to be independently associated with THN administration.

A sensitivity analysis of participants who injected drugs in the past year, had access to THN, and witnessed an overdose (n = 236) yielded comparable results (results not shown) with the exception of a few differences. Perception of a high prevalence of fentanyl in heroin

**Table 2. Multivariable model of correlates of access and administration take-home naloxone (THN) among people who use opioids in Baltimore, MD, USA.**

| | THN Access (N = 577) | | THN Administration (n = 345) | |
|---|---|---|---|---|
| | aOR (95% CI) | *P* value | aOR (95% CI) | *P* value |
| **Socio-Demographic Characteristics** | | | | |
| Age (ref: 21-39years) | | | | |
| 40–49 | 0.86 (0.48–1.55) | 0.612 | 0.55 (0.28–1.07) | 0.080 |
| 50–55 | 1.00 (0.55–1.81) | 0.992 | 0.75 (0.36–1.54) | 0.429 |
| 56+ | 0.70 (0.38–1.29) | 0.258 | 0.66 (0.31–1.40) | 0.276 |
| Male | 0.78 (0.50–1.19) | 0.249 | 0.96 (0.55–1.70) | 0.901 |
| High school/GED+ | 1.68 (1.13–2.50) | 0.010 | 1.37 (0.80–2.36) | 0.254 |
| Homeless | - - - | - - - | 1.82 (1.08–3.07) | 0.024 |
| **THN Availability** | | | | |
| Carry THN Often/Always | - - - | - - - | 3.47 (1.99–6.06) | < .001 |
| **Drug Use & Overdose History** | | | | |
| Injected in past 12mo | - - - | - - - | 2.61 (1.43–4.76) | 0.002 |
| Ever overdosed | 2.68 (1.79–4.01) | <0.001 | 0.61 (0.33–1.12) | 0.110 |
| No. of overdoses witnessed (ref: 1–2) | | | | |
| 3–4 | - - - | - - - | 2.02 (0.98–4.16) | 0.057 |
| 5–9 | - - - | - - - | 5.18 (2.22–12.07) | < .001 |
| 10+ | - - - | - - - | 5.73 (2.79–11.75) | < .001 |
| **Perceived Barriers of Responding** | | | | |
| Perception of insufficient overdose training | 0.43 (0.28–0.68) | <0.001 | 0.56 (0.32–0.96) | 0.034 |
| Perception that Narcan recipient will become aggressive | 0.55 (0.35–0.85) | 0.007 | 0.54 (0.28–1.04) | 0.064 |
| Perception that police will threaten people at overdose scene | 0.68 (0.46–1.00) | 0.052 | - - - | - - - |
| **Health Services Access and Engagement** | | | | |
| Medication-Assisted Treatment | 3.87 (2.59–5.78) | <0.001 | - - - | - - - |

- - -variable not included in the model

was associated with higher odds of administering THN (aOR: 2.33, 95% CI:1.23–4.42). Further, THN use was significantly reduced among those who reported fear that a person would be aggressive after administration of THN (aOR: 0.37, 95% CI:0.17–0.82). Additionally, among participants who injected drugs, homelessness, and perceptions of insufficient overdose response training were no longer statistically significantly associated with administering THN.

## Discussion

Among a community-based sample of PWUO in Baltimore, we identified several correlates of having access to THN and using it to respond to a witnessed overdose. We found that personally experiencing an overdose and current enrollment in MAT were facilitators of accessing

THN. THN access was reduced among PWUO who perceived that they had insufficient overdose response training, feared that a person would become aggressive after an overdose, and felt that police would threaten them with arrest or charges at an overdose scene. Consistently carrying THN and witnessing more overdoses were linked to use of THN. THN administration was lower among PWUO who perceived that they were insufficiently trained to respond to an overdose and feared that a THN recipient would become aggressive.

To the best of our knowledge, this is the first study to examine correlates of access to and administration of THN by PWUO in a large community-based sample. In contrast to Kenney et al. who found that people with personal overdose experience within an addiction treatment-based sample were more likely to have administered THN, personal overdose experience in this study was associated with access to THN but not with THN administration [18]. Corroborating the findings of Kenney and colleagues, our results suggest that recent injection drug use is strongly associated with administering THN [18].

The current study also identified that access to THN was significantly higher among people receiving MAT. This likely reflects engagement with the health care system as a facilitator for accessing THN. These data highlight the need to develop strategies to extend THN distribution to PWUO who are not engaged with substance use treatment services. Possible strategies could include training PWUO as peer mentors to distribute THN and provide overdose response training to their social networks. Detention facilities and social services accessed by PWUO are other key potential THN distribution points. Although drug treatment was associated with access, it was not associated with the use of THN. This lack of association may be a function of some drug treatment programs providing THN, but individuals in treatment may be less likely to witness an overdose and/or because they receive insufficient training to respond. Syringe service programs, drug treatment, and peer outreach could additionally serve as opportunities to diffuse information about the high proportion of fentanyl in heroin and associated elevated risk of overdose [24,25].

While engagement with services that provide THN influence the ability to access THN, individual factors may influence participants desire to access THN. A quarter of study participants expressed concern about overdose victims becoming aggressive after being revived with THN, which was associated with lower access to and administration of THN. An aggressive response post-revival may be the result of an overdose victim going into withdrawal or being disoriented [22,26]. The principles of trauma-informed care suggest that responders should supportively and clearly provide information about the status of the situation to the overdose victim after they have been revived [27]. THN training programs and outreach workers should provide training in de-escalation techniques and discuss the development of safety plans to further mitigate fears about responding to overdose [28]. Training programs can also emphasize the positive aspects of administering THN and reward individuals in the community who administer THN and hence save community members lives. Additionally, agitation post revival may be influenced by mode of THN administration. A randomized trial of intranasal versus intramuscular naloxone found that patients who received intramuscular naloxone exhibited higher rates of agitation (13%) compared to the intranasal naloxone group (2%) [29]. These findings may be due to the increased effectiveness of naloxone administered intramuscularly [6]. Pharmacological research is needed to determine dosing for opioid antidote medications to effectively respond to potent, fast-acting synthetic opioids while mitigating the risk of antidote-induced withdrawal [30].

The association between perceptions of police behaviors and receiving THN is important to note. In our analysis, we found perceived police threat was associated with lower access to THN among PWUO and among participants who injected drugs in the past year. Despite the passing of the Good Samaritan Law in Maryland, which protects anyone seeking medical assistance for an overdose victim from arrest and prosecution, half of PWUO perceived that police would threaten

people at an overdose occurrence with charges or arrest. These data suggest that attempts to use criminal justice approaches at overdose event scenes may unintentionally lead to reduced effectiveness of programs aimed at increasing access to and use of THN for fatal overdose prevention. Police threats at an overdose event may be particularly deleterious for people who inject drugs as they may be carrying drug-related paraphernalia. However, people who inject drugs are at high risk for overdose and are likely to witness overdose events, highlighting that they are key to engage in overdose prevention initiatives by removing barriers such as fear of police threat [31].

Carrying THN was one of the strongest correlates of THN administration identified in our study. This finding suggests the value of a clear message that encourages PWUO to always carry THN. Further research should explore ways to facilitate carrying THN, such as the provision of carrying cases for THN kits. Additionally, potential barriers to carrying THN, such as drug use stigma, warrant further study [19].

Our results also highlighted a need to improve THN trainings. Half of the participants who had access to THN and who witnessed an overdose agreed or strongly agreed with the statement that they needed more training to respond to an overdose event. These findings suggest a need for trainings that focus on increasing confidence to administer THN. Additionally, booster trainings and use of diverse education strategies may be necessary to reinforce skills beyond those provided during a single training session [32].

Study findings should be interpreted in the context of a few limitations. The cross-sectional study design limits our ability to determine temporality. Thus, the association of perceived need for overdose training and receiving THN could reflect that those with access were more likely to be trained or that those who felt that they did not have sufficient training did not seek THN. Third, this study was restricted to PWUO with a history of injecting drugs. Study findings may not be generalizable to PWUO who do not inject drugs. Generalizability may also be influenced by the older age of the study population which may not be representative of findings among younger PWUO. A final limitation is our study's definition of access to THN. Participants may have received a prescription but not have obtained THN, or they may have received THN previously but no longer have it. Future research should examine structural and individual-level factors associated with obtaining THN, among those with a prescription for THN. Additionally, an examination of how PWUO maintain a consistent supply of THN after use/loss merits further research.

The current study suggests that THN distribution programs in Baltimore have reached many individuals with a history of opioid and injection drug use, with just under half of those with access having administered THN. However, the rates of fatal overdose in Baltimore, and in many other cities, continue to rise [33–35]. Hence, there is a need for more aggressive distribution of THN beyond drug treatment settings. THN trainings should seek to increase confidence to administer THN as well as ensure responders' safety and comfort in responding. In addition, there is a need to change norms to encourage PWUO always to carry THN as well as how police interact with overdose victims and witnesses.

## Supporting information

**S1 Dataset. This is the supporting dataset.** Narcan_Data_PlosOne_6.13.19.
(XLSX)

## Acknowledgments

We thank the study participants who supported this work as well as Roeina Love, Tonya Johnson, Denise Mitchell, Charles Moore, Marlesha Bates, Abby Winiker and Joanne Jenkins for their assistance and support in data collection.

## Author Contributions

**Conceptualization:** Lauren Dayton, Rachel E. Gicquelais, Carl Latkin.

**Data curation:** Karin Tobin, Carl Latkin.

**Formal analysis:** Lauren Dayton, Rachel E. Gicquelais, Carl Latkin.

**Funding acquisition:** Karin Tobin, Carl Latkin.

**Investigation:** Lauren Dayton, Carl Latkin.

**Methodology:** Lauren Dayton, Rachel E. Gicquelais, Xiangrong Kong, Carl Latkin.

**Project administration:** Karin Tobin, Melissa Davey-Rothwell.

**Writing – original draft:** Lauren Dayton, Carl Latkin.

**Writing – review & editing:** Lauren Dayton, Rachel E. Gicquelais, Karin Tobin, Melissa Davey-Rothwell, Oluwaseun Falade-Nwulia, Xiangrong Kong, Michael Fingerhood, Abenaa A. Jones, Carl Latkin.

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
