## [Decision Letter · Decision Letter 0]

8 Aug 2019

PONE-D-19-16811

More than just access: Who obtains and who administers take-home naloxone in Baltimore, MD

PLOS ONE

Dear Ms. Dayton,

Thank you for submitting your manuscript to PLOS ONE. After careful consideration, we feel that it has merit but does not fully meet PLOS ONE’s publication criteria as it currently stands. Therefore, we invite you to submit a revised version of the manuscript that addresses the points raised during the review process.

We would appreciate receiving your revised manuscript by Sep 22 2019 11:59PM. To enhance the reproducibility of your results, we recommend that if applicable you deposit your laboratory protocols in protocols.io, where a protocol can be assigned its own identifier (DOI) such that it can be cited independently in the future. For instructions see: http://journals.plos.org/plosone/s/submission-guidelines#loc-laboratory-protocols

We look forward to receiving your revised manuscript.

Kind regards,

Bronwyn Myers

Academic Editor

PLOS ONE

Journal Requirements:

Additional Editor Comments (if provided):

Thank you for submitting this manuscript to PlOS One for consideration. We have now received two reviewer reports - both reviewers are enthusiastic about the manuscript and the relevance of the topic for the special issue. While noting that the manuscript is well-written, they do provide some recommendations for how to strengthen the manuscript even further while acknowledging a few additional limitations (see below). I look forward to receiving a revised manuscript that addresses these requests for minor revisions.

Reviewers' comments:

Reviewer's Responses to Questions

**Comments to the Author**

1. Is the manuscript technically sound, and do the data support the conclusions?

Reviewer #1: Yes

Reviewer #2: Yes

2. Has the statistical analysis been performed appropriately and rigorously? 

Reviewer #1: Yes

Reviewer #2: Yes

3. Have the authors made all data underlying the findings in their manuscript fully available?

Reviewer #1: Yes

Reviewer #2: Yes

4. Is the manuscript presented in an intelligible fashion and written in standard English?

Reviewer #1: Yes

Reviewer #2: Yes

5. Review Comments to the Author

Reviewer #1: The authors of this manuscript utilized data from a randomized clinical trial conducted in Baltimore, MD to investigate factors associated with obtaining and utilizing naloxone to reverse opioid overdose. This topic is of critical importance since take-home naloxone prevents fatal opioid overdoses if administered in a timely manner. There is strong evidence indicating that the most effective means of reversing opioid overdose with naloxone is to make it available to people who use drugs and are most likely to witness overdose. The authors clearly define their study aims and propose public health interventions to address their findings. Several fairly minor clarifying questions are listed below to improve this strong manuscript.

1) A stated implication of these findings is the need for more aggressive distribution of THN beyond drug treatment settings. Understanding where people access THN in this community is an important priority, particularly for people not engaged in care, and may give insight into how to expand access. Did the study assess where THN was accessed to add to these findings?

2) On a related note, I continued to consider the definition of and use of the term THN “access” throughout the manuscript. As the title itself suggests, it’s “More than just access: it is who obtains…” This distinction between lack of access (i.e., due to structural barriers or lack of opportunity to access it) vs. not obtaining THN despite access are two different outcomes and could be differentiated more clearly throughout the manuscript. The factors associated with decreased THN “access” – the fear that a person may become aggressive, fear of police threats, and insufficient overdose training – all reflect factors that would be associated with a lower likelihood of obtaining THN rather than factors that would affect likelihood of access. The authors state the limitations of their assessment of access (i.e., “have you ever been prescribed or received a kit containing Narcan?”) and while it may not be possible to distinguish between access to THN vs. obtaining THN given how it was assessed, I would still suggest re-considering the use of the term access for clarity.

3) The study focuses on a population of people who use opioids and have a history of injecting drugs. An additional limitation that should be noted is the specification of having injected drugs and findings possibly not being generalizable to PWUO who do not inject. This could be particularly relevant to those people who use prescription opioids who as the authors note in the introduction are less likely to have someone present during an overdose.

Minor:

1) Though I feel like it is implied that the authors are referring to intranasal naloxone throughout the paper, it would be helpful to explicitly state this in the introduction. In the context of the question of participants feeling that they have insufficient training, it would also be helpful for the reader to have a brief explanation of what naloxone training entails.

2) Though this may not need to be specifically described as a limitation, as a reader I’m curious why so few network members were recruited by index participants in the clinical trial. Is there any reason to think that those people who responded to outreach and advertisement to participate in the study may not be representative of PWUO in Baltimore City?

3) Please provide justification for dichotomizing responses to the three questions related to perceived barriers to assist in the event of an overdose.

4) P. 14 – “our results suggest that recent injection drug use and having witnessed an overdose are strongly associated with administering THN” – please clarify this statement, as all individuals included in the analyses predicting administration of THN were required to have witnessed an overdose.

Reviewer #2: This is a concise, well written description of access to and use of THN naloxone among over 500 at risk individuals in Baltimore, MD. The authors find important correlates and make appropriate recommendations. It would have been interesting to see if there were any associations of actually witnessing a fatality from an overdose. The associations might have been even stronger.

The advanced age of the study population may limit generalizability to younger populations and that should be noted.

6. PLOS authors have the option to publish the peer review history of their article (what does this mean?). If published, this will include your full peer review and any attached files.

Reviewer #1: Yes: Jessica Magidson

Reviewer #2: Yes: Josiah D. Rich, MD, MPH

---

## [Author Response · Author response to Decision Letter 0]

17 Sep 2019

We thank the academic editor and reviewers for their thoughtful review of our paper “More than just availability: Who has access and who administers take-home naloxone in Baltimore, MD.” We appreciate the suggestions and comments provided by reviewers and have responded to each point raised below.

EDITOR COMMENTS:

Thank you for submitting this manuscript to PlOS One for consideration. We have now received two reviewer reports - both reviewers are enthusiastic about the manuscript and the relevance of the topic for the special issue. While noting that the manuscript is well-written, they do provide some recommendations for how to strengthen the manuscript even further while acknowledging a few additional limitations (see below). I look forward to receiving a revised manuscript that addresses these requests for minor revisions.

Thank you for your review of the manuscript. As noted in the sections below, we have incorporated the suggestions of the reviewers. 

REVIEWERS COMMENTS: 

Reviewer #1: The authors of this manuscript utilized data from a randomized clinical trial conducted in Baltimore, MD to investigate factors associated with obtaining and utilizing naloxone to reverse opioid overdose. This topic is of critical importance since take-home naloxone prevents fatal opioid overdoses if administered in a timely manner. There is strong evidence indicating that the most effective means of reversing opioid overdose with naloxone is to make it available to people who use drugs and are most likely to witness overdose. The authors clearly define their study aims and propose public health interventions to address their findings. Several fairly minor clarifying questions are listed below to improve this strong manuscript.

1) A stated implication of these findings is the need for more aggressive distribution of THN beyond drug treatment settings. Understanding where people access THN in this community is an important priority, particularly for people not engaged in care, and may give insight into how to expand access. Did the study assess where THN was accessed to add to these findings?

We agree that understanding where people access take-home naloxone (THN) provides an important addition to this paper. The study assessed where participants received training to use Narcan, among those who stated that they had been trained to use Narcan (n=338). Study data found that participants were primarily trained through community outreach programs (36%), recovery programs (34%), and clinics (27%). The minority of participants reported being trained through detention centers (2%), through friends (1%), or other avenues (1%). These study findings are now incorporated into the results section. 

2) On a related note, I continued to consider the definition of and use of the term THN “access” throughout the manuscript. As the title itself suggests, it’s “More than just access: it is who obtains…” This distinction between lack of access (i.e., due to structural barriers or lack of opportunity to access it) vs. not obtaining THN despite access are two different outcomes and could be differentiated more clearly throughout the manuscript. The factors associated with decreased THN “access” – the fear that a person may become aggressive, fear of police threats, and insufficient overdose training – all reflect factors that would be associated with a lower likelihood of obtaining THN rather than factors that would affect likelihood of access. The authors state the limitations of their assessment of access (i.e., “have you ever been prescribed or received a kit containing Narcan?”) and while it may not be possible to distinguish between access to THN vs. obtaining THN given how it was assessed, I would still suggest re-considering the use of the term access for clarity.

In this paper we have defined access to THN as having received a kit of THN and/or a prescription for THN. While there is no standard definition of access, this definition is aligned with the WHO and NIDA definitions of access (NIDA, 2018, WHO, 2014). We agree with the reviewers’ comments that there is a difference between obtaining THN versus having a prescription for THN. We have therefore revised the manuscript in several ways. First, we amended the title (“More than just availability: Who has access and who administers take-home naloxone in Baltimore, MD”), which removed the word “obtain” to clarify that we did not only examine naloxone possession. Second, we delineate that structural factors affect one’s ability to access THN and individual-level factors may affect one’s desire to access THN in the discussion. We also added a recommendation that future research focus on PWUO who have a prescription for THN and examine factors that influence obtaining THN. As well as suggest that future research examine how people who use opioids maintain a consistent supply after use/loss. 

NIDA. (2018, June 8). Medications to Treat Opioid Use Disorder. Retrieved from https://www.drugabuse.gov/publications/research-reports/medications-to-treat-opioid-use-disorder on 2019, September 1

World Health Organization (WHO). Substance use: Community management of opioid overdose. 2014.

3) The study focuses on a population of people who use opioids and have a history of injecting drugs. An additional limitation that should be noted is the specification of having injected drugs and findings possibly not being generalizable to PWUO who do not inject. This could be particularly relevant to those people who use prescription opioids who as the authors note in the introduction are less likely to have someone present during an overdose.

We now include the limitation that study findings may not be generalizable to PWUO who do not inject drugs. 

Minor:

1) Though I feel like it is implied that the authors are referring to intranasal naloxone throughout the paper, it would be helpful to explicitly state this in the introduction. In the context of the question of participants feeling that they have insufficient training, it would also be helpful for the reader to have a brief explanation of what naloxone training entails.

We have included a description of available formulations of THN and mentioned that the core component of all programs is to enable the effective administration of THN in the management of an opioid overdose in the introduction. In general, intranasal naloxone appears to be more frequently available but due to price injectable naloxone is still distributed. 

2) Though this may not need to be specifically described as a limitation, as a reader I’m curious why so few network members were recruited by index participants in the clinical trial. Is there any reason to think that those people who responded to outreach and advertisement to participate in the study may not be representative of PWUO in Baltimore City?

The clinical trial did not require index participants to recruit a network member in hopes of boosting enrollment among those not wanting to enroll a network. The low number of network members may also be due to low remuneration for recruiting network members and the requirement that the network members have certain attributes. Network enrollment criteria included being 18 or older and did one of the following: 1) had sex with the index in the past 6 months; 2) injected drugs in the past 6 months; or 3) used drugs with client in the past 6 months. Though it is difficult to predict how networks differed from index participants or whether the PWUO in this study represented Baltimore City PWUO more generally, our recruitment at shelters may under-represent middle class PWUO or those who do not keep a primary residence in the city, but rather go between Baltimore city and county lines, which has become more common practice in the current opioid crisis.

3) Please provide justification for dichotomizing responses to the three questions related to perceived barriers to assist in the event of an overdose.

The three questions related to perceived barriers to assist at an overdose included perceptions of: insufficient training, that the THN recipient will become aggressive, and that police will threaten people at an overdose scene. These three questions were collected as categorical variables. Based on the distribution and extensive exploratory analyses, we chose to present the dichotomized results in order to enhance interpretability of study findings as the goal of this study was to inform programmatic design. Bivariate sensitivity analyses suggested that dichotomized and categorical study findings were consistent.

4) P. 14 – “our results suggest that recent injection drug use and having witnessed an overdose are strongly associated with administering THN” – please clarify this statement, as all individuals included in the analyses predicting administration of THN were required to have witnessed an overdose.

This statement has been revised to read: “Corroborating the findings of Kenney and colleagues, our results suggest that recent injection drug use is strongly associated with administering THN.”

Reviewer #2: 

1) This is a concise, well written description of access to and use of THN naloxone among over 500 at risk individuals in Baltimore, MD. The authors find important correlates and make appropriate recommendations. It would have been interesting to see if there were any associations of actually witnessing a fatality from an overdose. The associations might have been even stronger.

We agree with the reviewer that it would be interesting to assess the association between witnessing an overdose fatality with THN access and use. This data however was not collected in this dataset and therefore cannot be included in the present article. 

2) The advanced age of the study population may limit generalizability to younger populations and that should be noted.

We agree and now include older age of study population as a limitation to generalizability within the discussion section.

Again, we appreciate the input of the reviewers and feel that the article has been strengthened due to their contributions. Thank you for your consideration.

---

## [Decision Letter · Decision Letter 1]

21 Oct 2019

More than just availability: Who has access and who administers take-home naloxone in Baltimore, MD.

PONE-D-19-16811R1

Dear Dr. Dayton,

We are pleased to inform you that your manuscript has been judged scientifically suitable for publication and will be formally accepted for publication once it complies with all outstanding technical requirements.

With kind regards,

Bronwyn Myers

Academic Editor

PLOS ONE

Additional Editor Comments (optional):

Thank you for submitting this revised manuscript to PlOS One for review. I am pleased to inform you that the reviewers are happy with the changes you have made and this manuscript is now acceptable for publication.

Reviewers' comments:

Reviewer's Responses to Questions

**Comments to the Author**

1. If the authors have adequately addressed your comments raised in a previous round of review and you feel that this manuscript is now acceptable for publication, you may indicate that here to bypass the “Comments to the Author” section, enter your conflict of interest statement in the “Confidential to Editor” section, and submit your "Accept" recommendation.

Reviewer #1: All comments have been addressed

2. Is the manuscript technically sound, and do the data support the conclusions?

Reviewer #1: Yes

3. Has the statistical analysis been performed appropriately and rigorously? 

Reviewer #1: Yes

4. Have the authors made all data underlying the findings in their manuscript fully available?

Reviewer #1: Yes

5. Is the manuscript presented in an intelligible fashion and written in standard English?

Reviewer #1: Yes

6. Review Comments to the Author

Reviewer #1: The strengths of this manuscript, as described in the first review, have been enhanced in response to the previous recommendations. I expect that the dissemination of this work will promote further investigation on this important topic in the research community as well as potential changes to current public health practices in response to the authors’ recommendations (e.g. increased or enhanced THN training). Of note, this reviewer appreciates the access provided to the original data. The authors have been response to prior critiques, including the following:

1) Where people access THN: The authors incorporated their data on where participants reported obtaining THN. This information enhances the reader’s understanding of access to THN in the community and where resources might be best targeted in future interventions.

2) “Access” versus “Obtaining”: Though the authors took a different approach than was recommended, this reviewer believes that the choice to focus on the access to THN and distinguish this from obtaining THN substantially strengths the manuscript. The authors added a distinction between structural factors related to access and individual-level factors that may affect a person’s desire to obtain THN. While the data reported in this manuscript do not allow investigation of the factors that influence obtaining THN, the authors identify this as an important target for further research.

3) Focus on PWID: This reviewer feels that the inclusion of a limitation indicating that findings may not be generalizable to PWUD who do not inject drugs is an important addition. Explicit mention of this limitation may prompt further investigation by other researchers into how people who use prescription opioids without injecting access/use THN.

I believe the manuscript is suitable for publication in its current form. Only two minor comments remain that may further strengthen the manuscript:

1) Different formulations of THN: This reviewer appreciates the authors’ description of the different formulations/modes of administration of THN in the introduction. My question remains as to whether mode of administration further impacts use of THN? If the authors agree yet this data is not available, the manuscript could be further strengthened by brief mention of the limitation: lack of THN formulation data that could be associated with participant use of THN and/or perception of insufficient training.

2) Dichotomized participant responses: The authors provide a description of their decision to report participant responses of perceived barriers as dichotomous. Based on their mention of sensitivity analyses, I suggest including this information in the methods section (“Perceived Barriers of Responding,” pg. 7).

7. PLOS authors have the option to publish the peer review history of their article (what does this mean?). If published, this will include your full peer review and any attached files.

Reviewer #1: Yes: Jessica Magidson, Mary Kleinman

---

## [Editor Report · Acceptance letter]

30 Oct 2019

PONE-D-19-16811R1 

More than just availability: Who has access and who administers take-home naloxone in Baltimore, MD 

Dear Dr. Dayton:

I am pleased to inform you that your manuscript has been deemed suitable for publication in PLOS ONE. Congratulations! Your manuscript is now with our production department. 

With kind regards,

on behalf of

Dr. Bronwyn Myers 

Academic Editor

PLOS ONE